# Acculturation Associated with Sleep Duration, Sleep Quality, and Sleep Disorders at the US–Mexico Border

**DOI:** 10.3390/ijerph17197138

**Published:** 2020-09-29

**Authors:** Sadia B. Ghani, Marcos E. Delgadillo, Karla Granados, Ashley C. Okuagu, Pamela Alfonso-Miller, Orfeu M. Buxton, Sanjay R. Patel, John Ruiz, Sairam Parthasarathy, Patricia L. Haynes, Patricia Molina, Azizi Seixas, Natasha Williams, Girardin Jean-Louis, Michael A. Grandner

**Affiliations:** 1Sleep Health and Research Program and Department of Psychiatry, University of Arizona, Tucson, AZ 85724, USA; marcosde@email.arizona.edu (M.E.D.); kpaola.granados@gmail.com (K.G.); okuaguchisom@gmail.com (A.C.O.); pamela.alfonso@gmail.com (P.A.-M.); grandner@email.arizona.edu (M.A.G.); 2Centre for Sleep Research, Northumbria University, Newcastle-upon-Tyne NE18ST, UK; 3Department of Biobehavioral Health, The Pennsylvania State University, University Park, PA 16802, USA; omb104@psu.edu; 4Department of Medicine, University of Pittsburgh School of Medicine, Pittsburgh, PA 15261, USA; patelsr2@upmc.edu; 5Department of Psychology, University of Arizona, Tucson, AZ 85724, USA; johnruiz@arizona.edu; 6Center for Sleep Disorders and Division of Pulmonary, Allergy, Critical Care and Sleep Medicine, and Department of Internal Medicine, University of Arizona, Tucson, AZ 85724, USA; sparthasarathy@deptofmed.arizona.edu; 7Department of Health Promotion Sciences, University of Arizona Mel and Enid Zuckerman College of Public Health, Tucson, AZ 85724, USA; thaynes@email.arizona.edu; 8Senior Director, Mariposa Community Health Center, Nogales, AZ 85621, USA; pmolina@mariposachc.net; 9Department of Population Health, and Department of Psychiatry, NYU Langone Health, New York, NY 10016, USA; Azizi.Seixas@nyulangone.org (A.S.); Natasha.Williams2@nyulangone.org (N.W.); Girardin.Jean-Louis@nyulangone.org (G.J.-L.)

**Keywords:** acculturation, sleep, sleep disparities, Hispanic/Latino, health disparities, sleep duration, insomnia, medication use

## Abstract

Sleep disparities exist among Hispanics/Latinos, although little work has characterized individuals at the United States (US)–Mexico border, particularly as it relates to acculturation. This study examined the association of Anglo and Mexican acculturation to various facets of sleep health among those of Mexican descent at the US–Mexico border. Data were collected from *N* = 100 adults of Mexican descent in the city of Nogales, Arizona (AZ). Surveys were presented in English or Spanish. Acculturation was assessed with the Acculturation Scale for Mexican-Americans (ARSMA-II). Insomnia was assessed with the Insomnia Severity Index (ISI), sleepiness was assessed with the Epworth Sleepiness Scale (ESS), sleep apnea risk was assessed with the Multivariable Apnea Prediction (MAP) index, weekday and weekend sleep duration and efficiency were assessed with the Sleep Timing Questionnaire, sleep quality was assessed with the Pittsburgh Sleep Quality Index (PSQI), and sleep duration and sleep medication use were assessed with PSQI items. No associations were found between Mexican acculturation and any sleep outcomes in adjusted analyses. Anglo acculturation was associated with less weekend sleep duration and efficiency, worse insomnia severity and sleep quality, and more sleep apnea risk and sleep medication use. These results support the idea that sleep disparities may depend on the degree of acculturation, which should be considered in risk screening and interventions.

## 1. Introduction

The process of acculturation is multifaceted, focusing on cultural adaptation of groups and individuals that come into continuous contact with each other. The process of acculturation involves changes in areas of cultural, psychological, social, economic, and political factors [1]. For example, cultural changes may exert a positive influence on nutrition, but they may also exhibit negative ones as well [2].

Acculturation has also been associated with increased engagement in health risk behaviors that are highly prevalent in the United States. Acculturation was shown to be associated with unhealthy weight gain [3], lower rates of physical activity, and higher rates of fast-food consumption [4], as well as increased smoking and alcohol use [5].

Understanding the relationship of acculturation with the burden of health-related chronic conditions among individuals of Mexican descent is important due to the increasing numbers of these individuals in the United States and the existence of disparities in chronic diseases in this population. Mainly, research has shown that greater acculturation to the United States among Hispanic subgroups leads to multiple adverse health issues, which include increased body mass index [6] especially among Mexican-Americans [7,8].

Prior studies have examined acculturation and sleep, primarily among Hispanic/Latino populations. Mexican immigrants, compared to United States (US)-born Mexican-Americans, are about 40% more likely to be short sleepers after adjusting for demographic characteristics [9]. Mexico-born participants were significantly less likely than those born in the US to report short sleep [10] or difficulty falling asleep [11]. However, Mexican-Americans were more likely to report difficulties in maintaining sleep, early morning awakenings, nonrestorative sleep, and daytime sleepiness [11,12]. This suggests that factors such as culture may play a role in the maintenance of good sleep. However, it should be noted that most of this prior work did not assess acculturation with validated scales; rather, items measuring issues such as language use served as a proxy.

In a sample of around 300 women of Mexican descent in an urban Northern California community, positive associations between acculturation and self-reported sleep disturbances were reported [13]. Sleep difficulties were more often reported among women among women whose preferred language was English and also had earlier socialization to the United States. Similarly, Kachikis and Breitkopf [12] found that higher acculturation scores (on the basis of language) were associated with self-reported shorter sleep duration and lower sleep quality, among Hispanic/Latina women. Similar findings were seen when US-born Hispanic/Latina, Chinese, and Japanese immigrants were compared to their first-generation immigrant ethnic counterparts [14]. These findings are suggestive that first-generation immigrants who are less acculturated maintain better sleep, and greater acculturation is associated with worse sleep. On the other hand, in a study of US acculturation across first and second generations, working nonmanual labor jobs were associated with more general fatigue and shorter sleep duration [15,16].

Many prior studies are limited in that acculturation measures rely on single-item proxies (e.g., language). Although definitions of acculturation refer to changes in values, beliefs, attitudes, and behaviors as part of the process, it is not clear to what extent these elements are measured. Current instruments have varying dimensions but mainly capture changes in language use and proficiency [17]. Although changes in language may reflect a deeper acculturative change, they can also arise due to the necessity of language use and media preference. While language can be used as a measure of acculturation, it may fail to measure adoption of new values. Additional dimensions of acculturation should be taken into account such as length of stay in the United States, age of immigration person’s experience including food preference, media preference, and self-defined acculturation level [18,19]. This will help public health researchers better understand how the process of acculturation may influence health outcomes of individuals and their communities.

Another limitation of prior studies is that they focused on heterogeneous and often urban samples. Most sleep epidemiology among Hispanics/Latinos was mainly focused in urban settings [20,21] or largely made up of those in urban settings [22]. Less work has been documented in more vulnerable towns and rural communities. These communities are more insular and may face different challenges when compared to urban Hispanics/Latinos. Speaking Spanish language (always? In some contexts?) is identified as being a barrier to work in these regions. Nevertheless, health disparities at the border are an important public health concern, as identified by Healthy People 2020 [23].

Prior studies on sleep in Mexican-Americans revealed decreased rates of short sleep and insomnia symptoms. Hale and Rivero-Fuentes found that Mexican Immigrants had healthier sleep patterns than US-born adults of Mexican descent (i.e., Mexican-Americans), where Mexican-Americans were more likely to sleep less than 6.5 h daily [9]. Other studies showed that Mexican-Americans may have lower rates of short sleep [10], insomnia [11], and sleep apnea symptoms [11]. However, other studies suggested difficulties in sleep among Mexican-Americans, documenting problems in short sleep [24,25], insomnia [24,25,26], and sleep apnea risk [25]. This highlights the importance of clarifying measurement issues using standardized measures and examining the role of acculturation specifically, rather than just group membership more broadly.

The present study evaluated the role of acculturation related to sleep problems using a well-validated standardized acculturation rating scale among those of Mexican descent at the US–Mexico border. We hypothesize that, among participants, a greater degree of Anglo acculturation will be associated with poor sleep duration, insomnia, daytime sleepiness, and increased sleep medication use, whereas those of Mexican descent with less Anglo acculturation will have better sleep health.

## 2. Methods

### 2.1. Study

This was a pilot survey study of individuals living at the US–Mexico border. The sample was recruited as a convenience sample and, since this was a pilot study, no power calculations were performed, in accordance with statistical guidelines [26].

### 2.2. Participants

Participants included *N* = 100 adults of Mexican descent, recruited from the US–Mexico border, Nogales (Santa Cruz county, Arizona (AZ)). Of note, recruitment continued until *N* = 100 was reached. These individuals were recruited through in-person solicitations via booths set up outside of local shopping establishments, parks, and community buildings. Bilingual advertisements, flyers, and social media were also utilized. Inclusion criteria included (1) fluency in English or Spanish, (2) age over 18 years, (3) identifying as Mexican or Mexican-American, (4) having residence in Santa Cruz county, AZ, and (5) willingness to complete an extended survey battery. Exclusion criteria included (1) having a medical condition that would impede their ability to provide consent and/or participate, including (but not limited to) heart failure, type 1 diabetes, autoimmune conditions such as systemic lupus erythematosus, or uncontrolled serious mental illness (e.g., schizophrenia), (2) being under the age of 18 years, and (3) not residing in Santa Cruz county, AZ. This study was approved by the University of Arizona Institutional Review Board (protocol number 1608763580). All participants provided informed consent for inclusion and were compensated $20.

### 2.3. Measures

Acculturation was assessed using the Acculturation Rating Scale for Mexican-Americans II (ARSMA-II) [27]. ARSMA-II is a well-validated, standard measure of acculturation in this population [27]. It includes separate subscales for “Mexican acculturation” and “Anglo acculturation.” These subscales are independent of each other and range from 0–4, with higher scores indicating greater acculturation along each of those dimensions. In this way, an individual may score highly on either, both, or neither subscale. This approach allows for a separate investigation into the relative contribution of “Mexican” and “American” components of identity. Previous studies have shown that ARSMA-II is a useful measure for characterizing sleep in the context of acculturation [15]. ARSMA-II was initially developed and validated in both English and Spanish [27].

Insomnia was assessed using the Insomnia Severity Index (ISI) [28], a well-characterized, standard outcome measure for insomnia [28,29,30], which assesses the subjective severity of insomnia experiences. Seven items characterize perceived difficulties initiating, maintaining, and terminating sleep, satisfaction and worry about sleep, and elements of daytime dysfunction. The ISI, originally developed in English [28], was translated into Spanish and validated [31].

The Epworth Sleepiness Scale (ESS) is an eight-item measure of daytime sleep propensity, evaluating the degree of likelihood that an individual will fall asleep in a range of situations [32]. Items range from 0 (no chance of dozing) to 3 (high likelihood of dozing), and total scores range from 0–24, with higher values representing increased daytime sleepiness. The ESS is a standard clinical and research tool that is well validated [33]. The ESS was previously translated into Spanish [34].

Sleep quality was assessed using the Pittsburgh Sleep Quality Index (PSQI) [35]. The PSQI is a frequently used screening tool to broadly assess several dimensions of sleep overall good or poor sleep but is better at actually assessing sleepiness, sleep quality, insomnia symptoms, and sleep medication use [35,36]. It is a standard outcome measure in sleep research [37]. The PSQI computes an overall score on the basis of seven subscales, with scores over 5 considered as overall “poor” sleep quality. In addition to the overall score, the present study examined individual items of the PSQI. To assess general sleep duration, the item asking, “How many hours of actual sleep do you get at night (this may be different than the number of hours you spend in bed)?”, was used as a continuous variable. The item, “How often have you taken medicine (prescribed or over the counter) to help you sleep?”, was used to determine if individuals had a prior use of sleep medication. Responses that indicated any use in the past 30 days were coded as “yes.” The PSQI was previously translated into Spanish [38].

The Sleep Timing Questionnaire (STQ) was used to assess habitual computed weekday and weekend sleep duration and sleep efficiency [39]. This questionnaire was validated against a prospective sleep diary. The STQ asks individuals for times in and out of bed on weekdays and weekends and also asks questions related to typical sleep latency (SL) and wake after sleep onset (WASO). On the basis of these values, mean sleep time is calculated (using time in and out of bed to establish total time in bed (TIB), then calculating TIB–SL–WASO). Sleep efficiency is calculated as sleep time/TIB.

Sleep apnea risk was assessed using the Multivariable Apnea Prediction (MAP) index [40]. The MAP was developed to assess real-world sleep apnea risk on the basis of clinical variables (age, body mass index, and sex) and symptom reports (snoring, choking/gasping). On the basis of a previously published prediction equation, sleep apnea risk is characterized as percentage (0–100%) likelihood.

### 2.4. Translation of Measures into Spanish

Previously developed Spanish versions were used where possible, and English-only measures were translated for the purposes of this study. Guideline-based translation procedures were followed [41,42]. Specifically, an iterative process was followed for English-only measures (STQ and MAP): (1) the measure was translated by a Spanish-speaking physician, to ensure contextual accuracy; (2) the Spanish language measure was compared to the English measure by a bilingual non-researcher member of the community and revised; (3) the revised version was sent to a certified medical translator; (4) this version was evaluated by a focus group of *N* = 5 bilingual individuals from the community not involved with the research study and revised appropriately; (5) this version was then back-translated by an additional, blinded bilingual member of the community not involved in the study; (6) an additional focus group was asked to compare both the English and Spanish versions to check for compatibility; (7) lastly, the medical translator certified the accuracy of the final Spanish version. For previously translated measures (ISI, ESS, PSQI), steps 2 through 7 were still followed. Of note, this resulted in a new translation of the ISI, as items #4 and #6 were slightly changed in the process to accommodate regional language preferences.

### 2.5. Statistical Analyses

Linear regression analyses examined associations between acculturation (Mexican or Anglo, assessed as a continuous independent variable/predictor) and each continuous sleep-related outcome variable, including ISI score, ESS score, PSQI score, sleep duration (minutes), STQ weekday sleep time (minutes), STQ weekend sleep time (minutes), STQ weekday sleep efficiency (percent), STQ weekend sleep efficiency (percent), and MAP sleep apnea risk score (percent). Similarly, ordinal logistic regression analyses examined the likelihood of sleep medication use (yes vs. no), as predicted by Mexican and Anglo acculturation. All analyses were computed unadjusted and adjusted for age, sex, and education. Linear regression analyses examined unstandardized coefficients(B) and logistic regression analyses are reported as odds ratio (OR), along with 95% confidence intervals (CIs). A *p*-value below 0.05 was considered nominally statistically significant. All analyses were computed in STATA 14.0 (STATACORP, College Station, TX, USA).

## 3. Results

### 3.1. Characteristics of the Sample

Sample characteristics are reported in Table 1. The sample had a mean age of 36.5 years (SD = 19.1), and 47% were female. Among participants, 25% had a college degree. This is similar to US Census descriptions of the region, which estimated that the population is 52% female and 21% college graduates [43].

Mean Mexican acculturation was 2.9 points, and mean Anglo acculturation was 1.9 points. The mean score on the ISI was in the range indicative of mild insomnia, the mean score on the ESS was not indicative of excessive sleepiness, and the mean score on the PSQI was indicative of overall poor sleep quality. Sleep duration was self-reported as 6 h and 4 min on the PSQI, but 7 h and 16 min of computed weekday sleep duration, and 7 h and 49 min of computed weekend sleep duration was reported on the STQ. Sleep efficiency was estimated to be 88% on weekdays and 90% on weekends, indicating overall good sleep efficiency. Sleep apnea risk was moderate on average, and about one in five persons indicated a history of sleep medication use (see Table 1 for details).

### 3.2. Acculturation and Sleep-Related Outcomes

Results of regression analyses examining relationships between acculturation and sleep outcomes are reported in Table 2 for continuous outcomes. The odds ratios of sleep medication use are reported in Table 3, where medication use was the outcome variable/dependent variable with acculturation scores as the independent variable. In unadjusted analyses, Mexican acculturation was associated with greater sleep duration (but was no longer significant after adjustment). After adjusted analysis (for age, sex, and education), the degree of Mexican acculturation was not associated with any sleep outcomes.

Anglo acculturation, on the other hand, was associated with several outcomes. In unadjusted analyses, each point of Anglo acculturation on the ARSMA-II was associated with approximately one additional point on the ISI (1.3 points after covariate adjustment), one additional point on the PSQI (1.1 points after adjustment), 29 fewer minutes of sleep duration on the PSQI (33 min after adjustment), 41 fewer minutes of STQ weekend sleep duration (41 min after adjustment), four fewer percentage points of STQ weekend sleep efficiency (also 4% after adjustment), 7% increased sleep apnea risk (6% after adjustment), and 85% greater likelihood of prior use of sleep medications (132% greater likelihood after adjustment).

As a post hoc analysis, we examined whether any of these relationships depended on sex. Acculturation (Mexican and Anglo) by sex interactions were run for each outcome. The only significant interaction was seen for Mexican acculturation by sex for PSQI score (*p* < 0.05). When analyses were stratified by sex (still adjusted for age and education), greater Mexican acculturation was associated with lower PSQI score for men (B = −1.75; 95% CI −3.15 to −0.35; *p* = 0.015), but not women (B = 0.75; 95% CI −0.61 to 2.10; *p* > 0.05).

## 4. Discussion

The present study evaluated relationships between Mexican and Anglo acculturation and sleep variables among adults of Mexican descent at the US–Mexico border. Overall, the results showed that Mexican acculturation was not associated with sleep variables. Moreover, there was no association between acculturation and daytime sleepiness or weekday sleep duration or efficiency. Anglo acculturation was, however, associated with less sleep on weekends, lower weekend sleep efficiency, worse sleep quality, more insomnia, greater sleep medication use, and increased risk for sleep apnea.

The main finding of this study was that higher Anglo acculturation was associated with a variety of measures of poorer sleep health among individuals of Mexican descent at the US–Mexico border. This finding is consistent with previous work, which showed that those born in Mexico are less likely to report short sleep or difficulty falling asleep, when compared to those born in the US [10,11]. Subsequently, people of Mexican descent born in the US are more likely to have difficulties in sleep [11]. Similarly, US-born Mexican-Americans were more likely to be short sleepers when compared to Mexican immigrants [9]. Reasons for this association may be that those born in the US are likely to also engage in unhealthy behaviors including smoking, alcohol use, and lack of physical activity due to the possibility of having high stress levels as they adjust to living in the US. All these factors have been linked to poor sleep outcomes [9,44]. Other studies reported that Latinos in general (particularly Mexican-Americans) may be somewhat protected against sleep disparities seen in other groups [10,11,45]. It was suggested that this may be due to lack of acculturation, as being born in Mexico and speaking Spanish at home was protective of sleep quality in a nationally representative survey [10,11].

It is important to note that these previous studies did not investigate weekday–weekend differences. Anglo acculturation in this study was associated with less sleep on weekends. There are several possibilities for this. Those with more Anglo acculturation may be more likely to have employment schedules more consistent with typical weekday work hours, thus creating consistency in their schedules during the week. On weekends, they may tend to get less sleep possibly due to completing tasks left over from their week. It is also possible that, during the weekends, they are more engaged in social events such as hanging out with friends, going to dinner or clubs, binge-watching television, and spending time with family. Future studies should examine behavioral and time use patterns that may explain this.

Anglo acculturation in this population was associated with increased sleep medication use. This finding suggests that acculturation process has a positive effect on healthcare use and self-perceptions of health. More Anglo-accultured Hispanics/Latinos are more likely to use preventative services [46,47]. Possible explanations of this include higher rates of insurance coverage and improved healthcare access, enabling better management of chronic illnesses [48].

Findings for lower sleep duration and lower sleep efficiency were seen for weekend but not weekday sleep for those with greater degree of Anglo acculturation. This finding suggests that higher Anglo acculturation is associated with less sleep on weekends and, paradoxically, lower sleep efficiency. This finding highlights the importance of examining weekday and weekend sleep separately. These findings may also be related to different social behavior patterns, which should be explored in more detail in future studies.

There are several strengths of this study. The usage of ARSMA-II indicates significant progress in the sleep field. Compared to the simpler proxy measures (immigration status and language use) used in the past, ARSMA-II quantifies acculturation to Mexican and US culture along a continuous, two-dimensional scale. Thus, an individual can be high and/or low for both Mexican and US acculturation, providing a more informed approach to discussing acculturation in the public health sciences. Additionally, the participants in this study were selected from the US–Mexico border in Arizona, an understudied region. Most sleep epidemiology studies were conducted in urban settings and may miss challenges that those living in rural areas may face. Nogales, AZ is a suitable area for border health disparities research, particularly sleep epidemiology among Mexican-Americans.

The present study had several important limitations. First, the sample size was generally small. Despite being one of the largest studies of acculturation in this population using established measures, prevalence estimates and associations need to be replicated in a larger sample. Second, sleep measures were self-reported using validated instruments. Other details about sleep habits such as regularity of sleep schedule and naps, were not assessed. Prospective (e.g., sleep diary) and objective (e.g., actigraphy) measures were not available. Due to limited access to medical records, no information regarding the types of sleep medications was obtained. In addition, no previous diagnoses of medical conditions, specifically, sleep disorders, were available. For future studies, reliable medical records would be helpful to determine if sleep medication use or diagnosis of sleep disorders impacts the results obtained in this study.

Our study did not assess the impact of acculturative stress, i.e., the stress associated with adjusting to a new culture. We also did not measure the impact of stress among our study population, which may be a direct measure of risk for negative health outcomes associated with the process of acculturation. In the available literature, the impact of acculturative stress on acculturation varies, and the present study was not adequately powered to assess these complex relationships. For future studies, assessing for stress factors that may be associated with acculturation may assess its impact on sleep parameters, among Mexican-Americans.

The cross-sectional nature of analysis does not allow inference of causality in terms of causes of sleep outcomes among participants of Mexican descent with greater degree of Anglo acculturation. Lastly, there is concern for residual confounding that may exist due to unmeasured or poorly measures factors, such as chronic medical conditions, that tend to exist in most observational studies.

## 5. Conclusions

The present study identified associations between both Mexican and Anglo (i.e., American) acculturation and sleep disturbances among individuals of Mexican descent at the US–Mexico border. Overall, the study found that Mexican acculturation was not significantly associated with sleep, but Anglo acculturation in this group was associated with less sleep on weekends, worse sleep quality, more insomnia, greater sleep apnea risk, and more sleeping-pill use. Given the small sample size and cross-sectional nature, it did not identify mechanisms of these associations; these will need to be assessed in future work. Yet, these relationships aid in the identification of which individuals may be at increased risk for sleep problems. Of note, future research is needed to examine whether existing interventions (e.g., cognitive behavioral therapy for insomnia) are appropriate for this group, or if they need to be modified. Future studies should examine behavioral, social, and environmental mechanisms of these relationships, examine potential resilience factors in this community, and explore possible intervention strategies for mitigating some of these risks.

## Figures and Tables

**Table 1 ijerph-17-07138-t001:** Characteristics of the sample.

Variable	Category/Units	Distribution *
Age	Years	36.5 ± 19.1
Sex	Male	53%
	Female	47%
Education	College graduate	25%
	Some college	25%
	High school	23%
	Less than high school	27%
Acculturation	Mexican acculturation	2.90 ± 0.75
	Anglo acculturation	1.94 ± 0.94
ISI score	Points	9.14 ± 4.20
ESS score	Points	6.36 ± 4.33
PSQI score	Points	7.73 ± 3.47
PSQI sleep duration	Minutes	363.6 ± 105.0
STQ weekday sleep duration	Minutes	436 ± 144
STQ weekend sleep duration	Minutes	469 ± 162
STQ weekday sleep efficiency	Percent	88.4 ± 32.7
STQ weekend sleep efficiency	Percent	90.0 ± 13.9
MAP sleep apnea risk	Percent	30.2 ± 24.6
Sleep medication use	No	81%
	Yes	19%

* Values reported as percent or mean ± standard deviation (% or M ± SD); ISI, Insomnia Severity Index; ESS, Epworth Sleepiness Scale; PSQI, Pittsburgh Sleep Quality Index; STQ, Sleep Timing Questionnaire; MAP, Multivariable Apnea Prediction.

**Table 2 ijerph-17-07138-t002:** Association between acculturation and sleep characteristics among individuals of Mexican descent at the United States (US)–Mexico border.

Acculturation and Sleep		Unadjusted			Adjusted		
	Units	B	95% CI ^+^	*p*	B	95% CI ^+^	*p*
**Mexican Acculturation**							
Insomnia	ISI score	−0.26	(−1.38, 0.87)	0.651	−0.33	(−1.47, 0.81)	0.567
Sleepiness	ESS score	−0.08	(−1.24, 1.08)	0.894	−0.01	(−1.21, 1.19)	0.983
Sleep quality	PSQI score	−0.54	(−1.46, 0.39)	0.253	−0.44	(−1.36, 0.48)	0.348
Sleep duration	Minutes	28.44	(0.84. 56.04)	0.043	27.42	(−0.60, 55.44)	0.055
Weekday sleep time	Minutes	12.35	(−26.09, 50.79)	0.525	20.40	(−17.74, 58.54)	0.291
Weekend sleep time	Minutes	18.07	(−25.28, 61.41)	0.41	26.1	(−15.28, 67.48)	0.214
Weekday sleep efficiency	Percent	−0.43	(−9.26, 8.40)	0.923	0.50	(−8.64, 9.63)	0.914
Weekend sleep efficiency	Percent	2.73	(−0.98, 6.45)	0.147	3.45	(−0.19, 7.08)	0.063
Sleep apnea risk	Percent	3.40	(−3.16, 9.96)	0.307	2.95	(−2.08, 7.97)	0.248
**Anglo Acculturation**							
Insomnia	ISI score	1.06	(0.18, 1.93)	0.018	1.32	(0.44, 2.20)	0.004
Sleepiness	ESS score	0.33	(−0.59, 1.26)	0.476	0.38	(−0.59, 1.34)	0.441
Sleep quality	PSQI score	1.01	(0.30, 1.73)	0.006	1.13	(0.42, 1.84)	0.002
Sleep duration	Minutes	−28.68	(−50.40, −6.90)	0.01	−32.58	(−54.60, −10.50)	0.004
Weekday sleep time	Minutes	−5.62	(−36.37, 25.14)	0.718	−6.64	(−37.56, 24.29)	0.671
Weekend sleep time	Minutes	−40.57	(−74.36, −6.78)	0.019	−41.07	(−73.65, −8.49)	0.014
Weekday sleep efficiency	Percent	1.225	(−5.78, 8.23)	0.729	0.60	(−6.71, 7.91)	0.871
Weekend sleep efficiency	Percent	−3.98	(−6.85, −1.11)	0.007	−4.26	(−7.09, −1.43)	0.004
Sleep apnea risk	Percent	7.35	(2.29, 12.41)	0.005	5.57	(1.64, 9.49)	0.006

^+^ CI, confidence interval.

**Table 3 ijerph-17-07138-t003:** Association between acculturation and sleep medication use among individuals of Mexican descent at the US–Mexico border.

Acculturation and Sleep Medication	Unadjusted			Adjusted		
	OR ^+^	95% CI	*p*	OR ^+^	95% CI	*p*
Mexican Acculturation	0.95	(0.49, 1.83)	0.866	0.81	(0.40, 1.66)	0.571
Anglo Acculturation	1.85	(1.02, 3.35)	0.043	2.32	(1.16, 4.62)	0.017

^+^ OR, odds ratio.

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
