# Peer review of "Acculturation Associated with Sleep Duration, Sleep Quality, and Sleep Disorders at the US–Mexico Border"

_ijerph, 2020, doi:10.3390/ijerph17197138_

Round 1

Reviewer 1 Report

I have read with interest this article and I would like to congratulate to the authors for this interesting work. The manuscript presents experimental results.

I would like to ask some questions:

  1. In Table 1 are the data of continuous variables presented as mean with standard deviation? Data about age was clearly described as mean ± SD, but the values of other parameters are not clear. Please specify it clearly in all cases (for example in the legend of the table etc.)
  2. The authors wrote in line 212-213 (page 5) that Table 3 includes "the results of regression analysis examining relationships between acculturation and sleep for sleep medication use". I do not understand where the medication use is shown or explained in Table 3. Please describe it clearly or correct the Table with adequate information.
  3. Authors described in Table 1 that 19% of patients were on sleep medication. Do the authors have data about the type of the medications? Were these patients diagnosed with any type of sleep disorders before (for example insomnia, any type of parasomnias etc)? If the authors have information, please discuss it in the Results. Moreover, it could increase the statistical value of the results, if the authors rerun the statistical analysis after the exclusion of the patients with any type of sleep disorders and describe it in the Results, as an additional information.
  4. Authors should include power calculation in the Methods section. 
  5. Were there any differences in main outcomes between women and men in Anglo and Mexican acculturation?

The scientifically sound and methodological principles of the abstract and all paper are appropriate. The text is well structured.

Finally, I recommend the manuscript for major revision. The publication is not appropriate in the present form, it should be accepted after requested changes.

Reviewer 2 Report

Dear authors,

The present study entitled “Acculturation associated with sleep duration, sleep quality, and sleep disorders at the US-Mexico Border” is an interesting study.

However, I want to understand some points about the present study. In the Materials and Methods, the authors should describe the type of the present study before “Participants.” Provide the exclusion criteria.

Why you chose a hundred participants? Provide the power test of it. Provide the number of the ethical committee.

Overall, the methods should be improved.

Results Provide data about sleep medication. Discussion Prepare your text as only one section.

You should not separate from the text “Strengths” and “Limitations”

The sentence “Future studies should examine behavioral, social, and environmental mechanisms of these relationships, examine potential resilience factors in this community, and explore possible intervention strategies for mitigating some of these risks.” Should be included in your discussion.

Your conclusion should answer the aim of the study, no more than that. Keep safe. #Reviewer#

Reviewer 3 Report

In the present study, the authors have examined the association between both Mexican and Anglo acculturation among adults of Mexican descent at the US-Mexico border and have found that its the Anglo acculturation that shows association with poor sleep quality and other sleep problems. 

I think the authors have done a good job in writing this manuscript and have given abundant details in different sections.

I have a few minor concerns but other than those, I am fine with the quality of the manuscript. 

1) I fail to see how this study is going to be an effective way to help in  interventions without identifying the actual reasons causing these sleep problems. 

2) Do the subjects provide information about what they do and how stressful their lives are?Any way to put that information in this analysis?

3) How do these values compare to the Anglo group from a distant place and national averages?

Round 2

Reviewer 1 Report

I agree with the changes. The manuscript should be accepted in the revised form.